# Generative Skill Chaining: Long-Horizon Skill Planning with Diffusion Models

**Utkarsh A. Mishra, Shangjie Xue, Yongxin Chen, Danfei Xu**
Georgia Institute of Technology
Email:{umishra31,xsj,yongchen,danfei}@gatech.edu

**Abstract:** Long-horizon tasks, usually characterized by complex subtask dependencies, present a significant challenge in manipulation planning. Skill chaining is a practical approach to solving unseen tasks by combining learned skill priors. However, such methods are myopic if sequenced greedily and face scalability issues with search-based planning strategy. To address these challenges, we introduce Generative Skill Chaining (GSC), a probabilistic framework that learns skill-centric diffusion models and composes their learned distributions to generate long-horizon plans during inference. GSC samples from all skill models in parallel to efficiently solve unseen tasks while enforcing geometric constraints. We evaluate the method on various long-horizon tasks and demonstrate its capability in reasoning about action dependencies, constraint handling, and generalization, along with its ability to replan in the face of perturbations. We show results in simulation and on real robot to validate the efficiency and scalability of GSC, highlighting its potential for advancing long-horizon task planning. More details are available at: https://generative-skill-chaining.github.io/

**Keywords:** Manipulation Planning, Diffusion Models, Task and Motion Planning

## 1 Introduction

Long-horizon reasoning is crucial in solving real-world manipulation tasks that involve complex inter-step dependencies. An illustrative example is shown in Figure 1(bottom), where a robot must reason about the long-term effect of each action choice, such as the placement pose of the object and how to grasp and use the tool, in order to devise a plan that will satisfy various environment constraints and the final task goal (place object under rack). However, finding a valid solution often requires searching in a prohibitively large planning space that expands exponentially with the task length. Task and Motion Planning (TAMP) methods address such problems by jointly searching for a sequence of primitive skills (e.g., `pick`, `place`, and `push`) and their low-level control parameters. While effective, these methods require knowing the underlying system state and the kinodynamic models of the environment, making them less practical in real-world applications. This work seeks to develop a learning-based skill planning approach to tackle long-horizon manipulation problems.

Prior learning approaches that focus on long-horizon tasks often adopt the options framework [1, 2] and train meta-policies with primitive skill policies as their temporally-extended action space [3–8]. However, the resulting meta policies are *task-specific* and have limited generalizability beyond the training tasks. A number of recent works turn to *skill-level models* that can be composed to solve new tasks via test-time optimization [9–15]. Key to their successes are skill-chaining functions that can determine whether each parameterized skill can lead to states that satisfy the preconditions of the next skills in a plan, and eventually the success of the overall task. However, these methods are *discriminative*, meaning that they can only estimate the feasibility of a given plan and requires an exhaustive search process to solve a task. This bottleneck poses a severe scalability issue when dealing with increasingly complex and long skill sequences.

7th Conference on Robot Learning (CoRL 2023), Atlanta, USA.

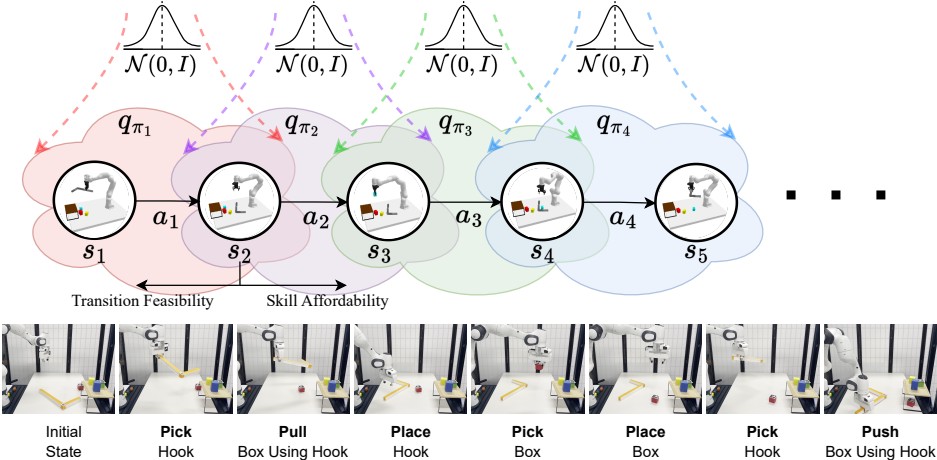

Figure 1: **(Top)** Generative Skill Chaining (GSC) aims to solve a long-horizon task for a given sequence of skills by using linear probabilistic chains to parallelly sample from a joint distribution of multiple skill-specific transitions $q_\pi(\mathbf{s}, \mathbf{a}, \mathbf{s}')$ learned using diffusion models. The framework implicitly considers transition feasibility and subsequent skill affordability while demonstrating flexible constraint-handling abilities. **(Bottom)** An example of a long-horizon TAMP problem composed of multiple skills. Such a task necessitates reasoning inter-dependencies between actions.

In this paper, we propose a *generative* and *compositional* framework that allows direct sampling of valid skill chains given a plan skeleton. Key to our method is skill-level probabilistic generative models that capture the joint distribution of precondition - skill parameter - effect of each skill. Sampling a valid skill chain boils down to, for each skill in a plan, conditionally generating skill parameters and post-condition states that satisfy the pre-condition of the next skill, constrained by the starting state and the final goal. The critical technical challenge is to ensure that the sequence of skill parameters is achievable from the initial state (*forward flow*) to satisfy the long-horizon goal (*backward flow*) and account for additional constraints.

To this end, we introduce **Generative Skill Chaining (GSC)**, a framework to train individual skill diffusion models and combine them according to given unseen task skeletons with arbitrary constraints at test time. Each skill is trained as an unconditional diffusion model, and the learned distributions are linearly chained to solve for an unseen long-horizon goal during evaluation. Further, we employ classifier-based guidance to satisfy any specified constraints. GSC brings a paradigm shift in the approaches used to solve TAMP problems to date and uses probabilistic models to establish compositionality and reason for long-horizon dependencies without being trained on any such task. We demonstrate the efficiency of GSC on three challenging manipulation task domains, explore its constraint handling advantages, and deploy a closed-loop version on a physical robot hardware to show generalization capabilities and robustness.

## 2 Related Work

**Task and Motion Planning.** Understanding inter-dependencies between sequential actions is a fundamental challenge in solving long-horizon problems. The key idea for solving them is to break the planning problem into a symbolically feasible sequence of smaller subtasks [1–3, 7, 8] and characterize their solutions with primitive actions or skills [4–6]. Such approaches rely on formulating fully-observable conditions and accurate system dynamics forecasting [16, 17] to realize the precondition of applying skill (affordability) and its effect respectively [18–24]. In this direction, logic-geometric programming [25, 26] and hierarchical [27] frameworks have solved for the symbolic feasibility [18, 28, 29] of a sequence of skills sufficient to reach a goal condition from the initial state. While such methods are exhaustive, their strong assumptions limit their practical applications and scalability. To overcome this, we opt for a learning-based framework [11, 30].

**Learning to solve long-horizon tasks.** Skill-chaining methods model pre- and post-conditions of pre-defined skills to search for feasible goal-reaching plans, but most methods have focused on single-task settings [9–15]. Recent methods have investigated composable skill models to learn multi-task planner [11, 27, 30]. However, such methods are discriminative and require exhaustive searches. Moreover, their auto-regressive nature leads to cascading errors and large exploration spaces as the tasks become long. Recently, Agia et al. [11] proposed a CEM-based skill-chaining strategy that maximizes the success of individual actions in the sequence by training individual skills as RL agents. Their method is still limited to trained policy (deterministic) priors, lack finding multi-modal solutions, and cannot account for additional planning constraints.

**Generative models for planning.** Generative models have been widely used for planning with Gaussian processes [31] and adversarial networks [32, 33]. With recent advancements in diffusion model-based generative strategies for planning in robotics, such approaches have been adopted for imitation learning [34–38] and offline reinforcement learning [39, 40] settings. Most relevant to this work are `Diffuser` [39], `Decision Diffuser` [40] and `Diffusion Policy` [37]. While they can synthesize sequences of "states" (and/or "actions") as plans and optionally account for constraints [40], such approaches still focus on capturing the distribution of training task solutions and cannot easily solve unseen new tasks. In this work, we introduce a compositional [41, 42] generative planning method that can flexibly combine *skill-level generative model* and generalize to new tasks and constraints introduced during inference time. The proposed framework is inspired by the recent work on generating high-quality extended image chains with parallel diffusion [43] and scalable diffusion models with transformers [44].

## 3 Preliminaries

**Problem formulation.** We formulate the skill chaining problem by considering a given symbolically feasible skeleton $\Phi_K = \{(\pi_1, o_1), (\pi_2, o_2), \ldots, (\pi_K, o_K)\}$ of skills from a pre-defined library $\pi_{1:K} \in \Pi$ and the set of objects $o$ on which the skill operates. Each skill $\pi \in \Pi$ is parameterized by a continuous set of feasible skill parameters $\mathbf{a} \in \mathcal{A}_\pi$ governing the desired motion while executing the skill from a state $\mathbf{s}$ in the state space $\mathcal{S}$. The goal is to find optimal skill parameters such that the skeleton is geometrically feasible and the effect of each skill satisfies the precondition of the following skills while the goal condition is satisfied.

**Environment setup.** We use expert policies to solve individual skills and collect *state-action-state* transitions between current ($\mathbf{s} \in \mathcal{S}$) and next state ($\mathbf{s}' \in \mathcal{S}$) resulting from the execution of a skill $\pi$ with parameters $\mathbf{a}_\pi \in \mathcal{A}_\pi$ by following the transition model $\mathcal{T}_\pi : \mathcal{S} \times \mathcal{A}_\pi \to \mathcal{S}$. Further, following previous work [11], we consider a selection of basic objects like a hook, a rack, and boxes of various dimensions to construct the environmental setup. The state of the environment consists of fully observable poses and sizes of each of the present objects.

**Diffusion models.** The diffusion model is a parameterized model $p_\theta(\mathbf{x}_0)$ that estimates an unknown distribution $q(\mathbf{x}_0)$ using the samples $\mathbf{x}_0 \sim q_0(\mathbf{x}_0)$. It consists of two diffusion processes: a forward noising and a reverse denoising process. The forward process progressively injects *i.i.d.* Gaussian noise to samples from $q_0(\mathbf{x}_0)$ and leads to a family of noised distributions $q_t(\mathbf{x}_t)$. The distribution of $\mathbf{x}_t$ conditioned on the clean data $\mathbf{x}_0$ is also a Gaussian: $q_{0t}(\mathbf{x}_t|\mathbf{x}_0) = \mathcal{N}(\mathbf{x}_t; \mathbf{x}_0, \sigma_t^2\mathbf{I})$, where $\sigma_t$ defines a fixed series of noise levels monotonically increasing w.r.t. forward diffusion $t$. The reverse denoising process recovers clean data by iteratively removing the added noise by a process represented as the following stochastic differential equation (SDE) [45–47]:

$$d\mathbf{x} = -2\dot{\sigma}_t\sigma_t\nabla_\mathbf{x}\log q_t(\mathbf{x}_t)dt + \sqrt{2\dot{\sigma}_t\sigma_t}d\mathbf{w} \tag{1}$$

where $\nabla_\mathbf{x}\log q_t(\mathbf{x}_t)$ is referred to as the score function of the noised distribution and $\mathbf{w}_t$ is a standard Wiener process. We follow DDPM [48] sampling strategy in continuous settings [49]. The score function allows recovery of the minimum mean squared error estimator of $\mathbf{x}_0$ given $\mathbf{x}_t$ [50, 51]:

$$\tilde{\mathbf{x}}_0 := \mathbb{E}[\mathbf{x}_0|\mathbf{x}_t] = \mathbf{x}_t + \sigma_t^2\nabla_{\mathbf{x}_t}\log q_t(\mathbf{x}_t), \tag{2}$$

where we can treat $\tilde{\mathbf{x}}_0$ as a "denoised" version of $\mathbf{x}_t$ at timestep $t$. In practice, the unknown score function is estimated using a neural network $\epsilon_\theta(\mathbf{x}_t, t)$ by minimizing the denoising score matching [49] objective $\mathbb{E}_{t,\mathbf{x}_0}[\lambda(t)\|\sigma_t \nabla_{\mathbf{x}_t} \log q_{0t}(\mathbf{x}_t|\mathbf{x}_0) - \epsilon_\theta(\mathbf{x}_t, t)\|^2]$ where $\lambda(t)$ is a time-dependent weight. Diffusion models are scalable, and the learned distributions represent all positive samples satisfying the distribution heuristic, thus multi-modal. Further, their simple probabilistic representation allows a wide range of flexible sampling strategies [47, 48, 52, 53] combined with constraint-handling abilities [40, 54–56].

## 4 Methodology

Generative Skill Chaining (GSC) offers a new paradigm for approaching long-horizon planning with a given skeleton of skills. The primary objective of GSC is to determine the optimal skill parameters for an unseen task skeleton, such that executing the plan achieves a long-horizon goal while satisfying task-specific constraints. It introduces probabilistic chaining of distributions of short-horizon transitions to sample from a long-horizon trajectory distribution. GSC uses skill-level diffusion models to represent each skill's joint distribution of precondition, control parameters, and effect. Further, the framework composes individual skills at inference time to form a sequence-level trajectory distribution, which can be sampled via parallel diffusion to generate feasible skill parameter sequences as planning solutions. This is different from the widely used auto-regressive heuristic-search-based approach [17, 57–59] used in prior works [11, 30].

We consider a given skeleton $\Phi$ of skills (and relevant objects) which satisfies the symbolic feasibility of the sequence in the environment. The primary goal is to generate the sequence of states and skill parameters (as shown in Figure 2(a)) such that the final state (here $\mathbf{s}_f \equiv \mathbf{s}^{(2)}$) satisfies a goal condition and leads to the successful execution of the last skill.

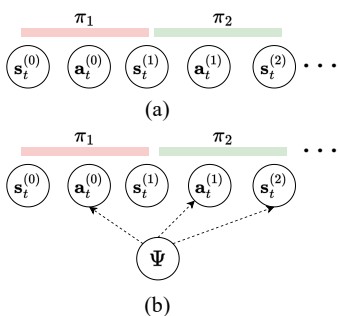

(a)

(b)

Figure 2: **(a)** A linear chain graph for a long sequence of transitions and **(b)** adding an additional constraint node.

**Action primitives as diffusion models.** We characterize the individual skills by the nature of *state-action-state* transitions observed while executing it in the environment. The operation of each skill $\pi$ can then be represented by an unconditional distribution $q_\pi(\mathbf{s}_t, \mathbf{a}_t, \mathbf{s}'_t)$. Such a representation simultaneously captures the skill policy $P_\pi$ and the transition dynamics distribution $T_\pi$ and ensures their consistency. For each skill $\pi$ in the skill library, we train a diffusion model score function $\epsilon_\pi(\mathbf{s}_t, \mathbf{a}_t, \mathbf{s}'_t, t)$ with transformer backbone as shown in Figure 3 (right) using provided per-skill demonstration data. We represent the set of objects of interest by an order which denotes the relevance of the objects in the scene w.r.t. the skill[1] [11]. We also denote the masked sampling score model for the states and action as $\epsilon_\pi(\mathbf{s}_t, t)$, $\epsilon_\pi(\mathbf{s}'_t, t)$ and $\epsilon_\pi(\mathbf{a}_t, t)$ respectively.

**Sequencing skill diffusion models.** To solve our objective of finding a sequence of suitable skill transitions which satisfies $\Phi$, an auto-regressive approach primarily used in prior works follows:

$$p_\Phi(\mathbf{s}^{(0:2)}, \mathbf{a}^{(0:1)}|\mathbf{s}^{(0)}) \equiv P_{\pi_1}(\mathbf{a}^{(0)}|\mathbf{s}^{(0)})\ T_{\pi_1}(\mathbf{s}^{(1)}|\mathbf{s}^{(0)}, \mathbf{a}^{(0)})\ P_{\pi_2}(\mathbf{a}^{(1)}|\mathbf{s}^{(1)})\ T_{\pi_2}(\mathbf{s}^{(2)}|\mathbf{s}^{(1)}, \mathbf{a}^{(1)})$$

However, such formulations are myopic and can only be rolled out in the forward direction without feedback from the final task goal. This limits long-horizon reasoning, and prior methods have leveraged random [30] or CEM-based rollouts [11] to sample from such a distribution. To overcome the above limitations, we transform the unconditional skill diffusion models into a forward and a backward conditional distribution, as

$$p_\Phi(\mathbf{s}^{(0:2)}, \mathbf{a}^{(0:1)}|\mathbf{s}^{(0)}) \propto q_{\pi_1}(\mathbf{s}^{(0)}, \mathbf{a}^{(0)}, \mathbf{s}^{(1)})q_{\pi_2}(\mathbf{a}^{(1)}, \mathbf{s}^{(2)}|\mathbf{s}^{(1)}) = \frac{q_{\pi_1}(\mathbf{s}^{(0)}, \mathbf{a}^{(0)}, \mathbf{s}^{(1)})q_{\pi_2}(\mathbf{s}^{(1)}, \mathbf{a}^{(1)}, \mathbf{s}^{(2)})}{q_{\pi_2}(\mathbf{s}^{(1)})}$$

---

[1]For example, if there is a hook (1), a box (2) and a rack (3) in the environment, then the object order corresponding to tasks are: (a) pick the box: $[2, 1, 3]$, (b) Place Box on Rack: $[2, 3, 1]$.

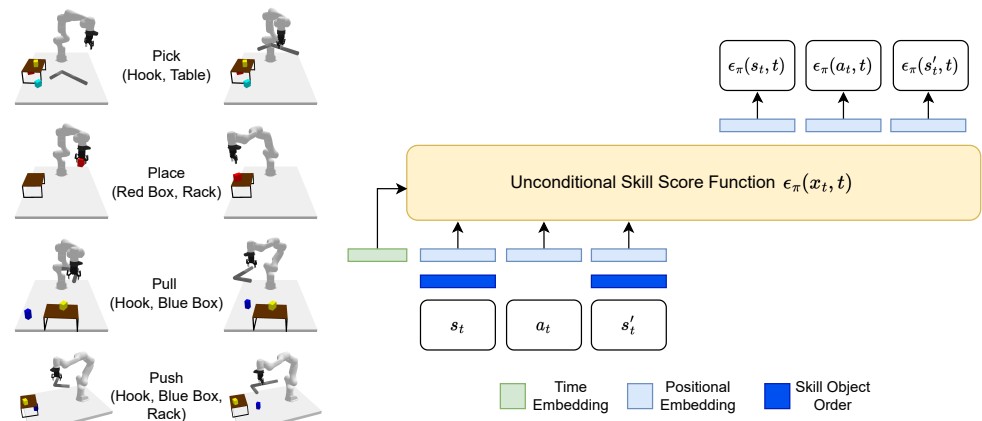

Figure 3: **Left** The primitive skills and their executions are shown with the objects of interest. **Right** Transformer-based skill diffusion model. We use the noisy *state-action-state* distribution $\mathbf{x}_t \sim \{\mathbf{s}_t, \mathbf{a}_t, \mathbf{s}'_t\}$ at diffusion step $t$ to obtain the corresponding $\epsilon_\theta$ during sampling. The skill object order depends on the objects of interest and is represented as a collection of one-hot vectors.

$$p_\Phi(\mathbf{s}^{(0:2)}, \mathbf{a}^{(0:1)}|\mathbf{s}^{(2)}) \propto q_{\pi_1}(\mathbf{s}^{(0)}, \mathbf{a}^{(0)}|\mathbf{s}^{(1)}) q_{\pi_2}(\mathbf{s}^{(1)}, \mathbf{a}^{(1)}, \mathbf{s}^{(2)}) = \frac{q_{\pi_1}(\mathbf{s}^{(0)}, \mathbf{a}^{(0)}, \mathbf{s}^{(1)}) q_{\pi_2}(\mathbf{s}^{(1)}, \mathbf{a}^{(1)}, \mathbf{s}^{(2)})}{q_{\pi_1}(\mathbf{s}^{(1)})}$$

In both equations above, the relations implicitly give rise to the notion of skill affordability and transition feasibility, i.e., the resulting state from the one skill must lie in the initial state distribution of the next skill and vice-versa. Now, if we transform the probabilities into their respective score functions ($\nabla_\mathbf{x} \log q(\mathbf{x})$) for a particular reverse diffusion sampling step $t$, we obtain:

$$\epsilon_\Phi(\mathbf{s}_t^{(0)}, \mathbf{a}_t^{(0)}, \mathbf{s}_t^{(1)}, \mathbf{a}_t^{(1)}, \mathbf{s}_t^{(2)}, t) = \epsilon_{\pi_1}(\mathbf{s}_t^{(0)}, \mathbf{a}_t^{(0)}, \mathbf{s}_t^{(1)}, t) + \epsilon_{\pi_2}(\mathbf{s}_t^{(1)}, \mathbf{a}_t^{(1)}, \mathbf{s}_t^{(2)}, t) - \epsilon_{\pi_2}(\mathbf{s}_t^{(1)}, t) \quad (3)$$

$$\epsilon_\Phi(\mathbf{s}_t^{(0)}, \mathbf{a}_t^{(0)}, \mathbf{s}_t^{(1)}, \mathbf{a}_t^{(1)}, \mathbf{s}_t^{(2)}, t) = \epsilon_{\pi_1}(\mathbf{s}_t^{(0)}, \mathbf{a}_t^{(0)}, \mathbf{s}_t^{(1)}, t) + \epsilon_{\pi_2}(\mathbf{s}_t^{(1)}, \mathbf{a}_t^{(1)}, \mathbf{s}_t^{(2)}, t) - \epsilon_{\pi_1}(\mathbf{s}_t^{(1)}, t) \quad (4)$$

respectively. Finally, we linearly combine the score functions from the forward and backward distributions weighted by a dependency factor $\gamma$:

$$\epsilon_\Phi(\mathbf{s}_t^{(1)}, t) = \gamma_{\pi_1}\, \epsilon_{\pi_1}(\mathbf{s}_t^{(1)}, t) + (1 - \gamma_{\pi_1})\, \epsilon_{\pi_2}(\mathbf{s}_t^{(1)}, t), \quad (5)$$

Here, $\gamma \in [0, 1]$ is a decision variable that balances the influence of the state in the transition of the skill w.r.t. the subsequent skill and the goal condition. This is an important aspect that governs the behavior of the skills in the sequence and the choice of their respective parameters.

**Classifier-based guidance for constraint satisfaction.** Besides the final task goal, constraints play an important role in governing the feasibility of actions in the environment and specifying task-specific conditions, such as maximizing/minimizing the distance between two objects in an intermediate or final state. Our diffusion model-based formation allows GSC to easily incorporate additional constraints as implicit (in-painting) or explicit (classifier-based) guidance. Here we present a flexible sampling strategy in the presence of several planning constraints. In principle, the additional constraints $\Psi$ can be appended as additional terms in the target sampling distribution:

$$p_{\Phi,\Psi}(\mathbf{s}^{(0:2)}) \propto p_\Phi(\mathbf{s}^{(0:2)}|\mathbf{s}^{(0)})\ h(\{\mathbf{s}, \mathbf{a}\}_\Psi)$$

where $h(\cdot)$ is the likelihood of the constraint acting on a set of state-action nodes given by $\{\mathbf{s}, \mathbf{a}\}_\Psi$. The corresponding diffusion score function with the added constraints becomes

$$\epsilon_{\Phi,\Psi}(\mathbf{s}_t^{(0)}, \mathbf{a}_t^{(0)}, \mathbf{s}_t^{(1)}, \mathbf{a}_t^{(1)}, \mathbf{s}_t^{(2)}, t) = \epsilon_\Phi(\mathbf{s}_t^{(0)}, \mathbf{a}_t^{(0)}, \mathbf{s}_t^{(1)}, \mathbf{a}_t^{(1)}, \mathbf{s}_t^{(2)}, t) + \epsilon_\Psi(\{\mathbf{s}_t, \mathbf{a}_t\}, t) \quad (6)$$

where $\epsilon_\Psi(\{\mathbf{s}_t, \mathbf{a}_t\}, t) \propto \nabla_{\{\mathbf{s}_t, \mathbf{a}_t\}} \log h(\{\mathbf{s}_t, \mathbf{a}_t\}_\Psi)$. Consider the example shown in Figure 2(b) where the constraint depends on the nodes $\mathbf{a}^{(0)}, \mathbf{a}^{(1)}$ and $\mathbf{s}^{(2)}$. Suppose the constraint is chosen to be a binary indicator (i.e. success = 1) of satisfaction and is defined for the denoised samples ($\mathbf{a}_0^{(0)}, \mathbf{a}_0^{(1)}$ and $\mathbf{s}_0^{(2)}$) at $t = 0$. In such a situation, the likelihood is defined as the exponential of the constraint satisfaction such that

$$h_\Psi(\mathbf{a}_0^{(0)}, \mathbf{a}_0^{(1)}, \mathbf{s}_0^{(2)}) = \exp\left[ -\alpha\left(1 - \Psi(\mathbf{a}_0^{(0)}, \mathbf{a}_0^{(1)}, \mathbf{s}_0^{(2)})\right)\right] \quad (7)$$

It is worth noting that while the constraint is a function of the denoised samples, the gradients must be calculated w.r.t. the noised samples. We calculate this by first obtaining the denoised sample $\tilde{\mathbf{x}}$ for the diffusion step $t$ from Equation 2 and then modifying the corresponding nodes in $\epsilon_\Phi(\mathbf{s}_t^{(0)}, \mathbf{a}_t^{(0)}, \mathbf{s}_t^{(1)}, \mathbf{a}_t^{(1)}, \mathbf{s}_t^{(2)}, t)$ based on the weight factor $\alpha$, as

$$\tilde{\epsilon}_\Phi(\mathbf{a}_t^{(0)}, \mathbf{a}_t^{(1)}, \mathbf{s}_t^{(2)}, t) = \epsilon_\Phi(\mathbf{a}_t^{(0)}, \mathbf{a}_t^{(1)}, \mathbf{s}_t^{(2)}, t) - \alpha \nabla_{\mathbf{a}_t^{(0)}, \mathbf{a}_t^{(1)}, \mathbf{s}_t^{(2)}} \left(1 - \Psi(\tilde{\mathbf{a}}_0^{(0)}, \tilde{\mathbf{a}}_0^{(1)}, \tilde{\mathbf{s}}_0^{(2)})\right) \quad (8)$$

**Summary.** To summarize, the proposed framework GSC is divided into three segments: (1) train individual skill diffusion models with the proposed architecture without any knowledge about other skills, (2) chain skill diffusion models according to an unseen task skeleton during inference using probabilistic linear chaining of the individually learned distributions with a dependency factor, and (3) incorporate classifier-based guidance for any unseen planning constraint added while inference. Following standard reverse denoising, we consider parallelly sampling from all individual models instead of one and hence our proposed approach is both task-skeleton and skeleton-length agnostic. Further, the dependency factor helps in making flexible design choices for satisfying the desired goal condition. While a constant value of $\gamma = 0.5$ is sufficient, it can be fine-tuned for every skill. In addition to the above, we also collect failure data to train a success probability prediction module $Q(\mathbf{s}, \mathbf{s}') : \mathcal{S} \times \mathcal{S} \to [0, 1]$ for each skill which is a measure of the successful execution of the skill given the current and the transitioned state. Such a model is used to consider the best parameter sequence from the sampled candidate solutions. We illustrate the overall algorithm in Appendix A.

## 5 Results

We conduct experiments to validate the efficacy of GSC in (1) long-horizon planning for unseen tasks of arbitrary lengths, (2) constraint handling and satisfaction, and (3) maximizing action-dependency horizon and finally (4) generalization to perturbations. First, we show the compositional and constraint-handling performance of GSC in a toy domain. Second, we evaluate the performance of the chaining trained skill diffusion models on nine standard TAMP tasks introduced by previous work [11]. These tasks encompass a wide range of skeleton lengths and challenge the method on various levels of long-horizon dependency. Finally, we discuss the response of GSC to perturbations, followed by the importance of dependency factor $\gamma$ in the success of GSC.

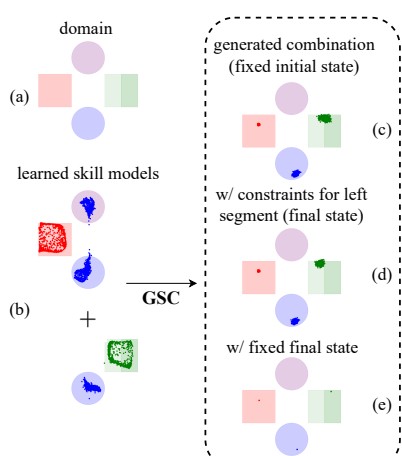

Figure 4: **Toy Domain:** We model four distributions of states and segment one of them into left and right segments. The figure illustrates diffusion model composition using GSC with fixed start state and unit-vector actions followed by the addition of soft and hard constraint guidance.

**Baselines and metrics.** In the context of skill chaining, we primarily consider search-based methods with CEM optimization strategy. Our main baselines are CEM methods with uniform priors (**Random CEM**) and learned policy priors (**STAP**). Further, to show improvement in performance as compared to training on task sequences and expecting generalization to new ones, we add **DAF**'s [30] performance following Agia et al. [11]. Another potential baseline is using diffusion for states only and inverse dynamics model for actions based on **Decision Diffuser** [40]. However, due to considerable distribution shift and cascading error from state predictions, such a method does not perform well (refer to Appendix B). The success rate of satisfying the goal condition in 100 random environment executions is used as the comparison metric[2].

**Toy domain.** The 2D domain consists of states as the $\mathbf{s} \triangleq (x, y)$ position of a point sample and the action, $\mathbf{a} = [r, \theta] \in \mathbb{R}^2$, as the direction of the unit vector ($\theta$) and magnitude ($r$). A transition is

---

[2]We consider the same results for baselines as demonstrated in previous work [11].

defined by $\mathbf{s}' = \mathbf{s} + [r\cos(\theta), r\sin(\theta)]$. To replicate the skill distributions, we create multiple state distributions Figure 4(a) and collect transition data between them to represent individual skills. We illustrate the performance of GSC in chaining trained skill models Figure 4(b) to sample transitions between sequence of distributions Figure 4(c) with fixed initial state and unit action. Finally, we validate the constraint satisfaction performance by first constraining final states to the left segment and second, keeping the goal state fixed Figure 4(d)(e), respectively. The success in the above tasks validates the framework's ability to compose skills and handle arbitrary constraints directly while evaluation. Further clarification for the toy domain is provided in Appendix C.

**Long-horizon manipulation.** We consider TAMP problems in a PyBullet [60] environment and follow the tasks proposed by STAP [11], namely: **(i) Hook Reach:** The hook is used to pull a box followed by executing other skills on the box. **(ii) Constrained Packing:** Multiple boxes should be placed on the rack so that all of them can be accommodated without interference. **(iii) Rearrangement Push:** A sequence of placement and push using a hook is executed to bring a box below the rack. The proposed three skeletons for each category above are solved using our method and compared against the previous work and their baselines w.r.t. the success rate is shown in Table 1. These tasks are unseen while training, have varying skeleton lengths, and demand reasoning long-horizon action dependencies. The performance of GSC implies that it is better than (or as good as) search-based methods, along with other advantageous abilities. The details about the target skeletons and the desired transitions are further explained in Appendix E.

**Imposing additional constraints.** In addition to the skeletons evaluated for the long-horizon problems and extending the experiments conducted for the toy domain, we impose certain planning constraints on the final state, intermediate states, and actions. This is done by adding an objective of maximizing the distance between all the "place" skill action parameters in the skeleton (this addition is still task agnostic) for the `constrained packing` task. The resulting sequence of states is shown in Figure 5. Further, we compare cumulative task completion and constraint satisfaction success rate with previous approaches in Table 2. This qualitatively and quantitatively demonstrates that the framework can handle such unseen constraints in test time.

Table 1: The success rate of the proposed GSC algorithm is shown and compared with relevant search-based baselines (CEM strategy). All results are calculated from 100 trials for each task.

| Methods | Hook Reach | | | Rearrangement Push | | |
|---|---|---|---|---|---|---|
| | Task 1 | Task 2 | Task 3 | Task 1 | Task 2 | Task 3 |
| Random CEM | 0.54 | 0.40 | 0.30 | 0.30 | 0.10 | 0.02 |
| DAF (Generalization) | 0.32 | 0.05 | 0.0 | 0.0 | 0.08 | 0.0 |
| STAP (Policy CEM) | 0.88 | 0.82 | 0.76 | 0.40 | 0.52 | 0.18 |
| GSC (Ours) | 0.84 | 0.84 | 0.76 | 0.68 | 0.60 | 0.18 |
| Task Length | 4 | 5 | 5 | 4 | 6 | 8 |

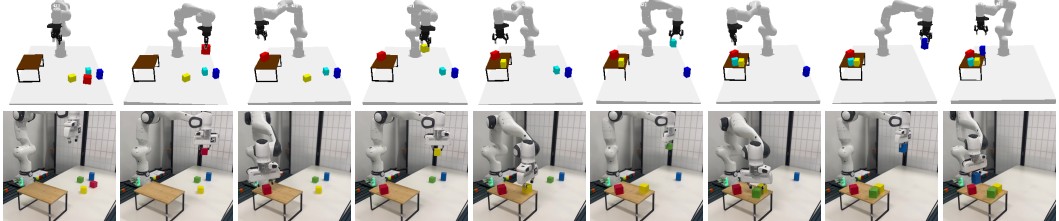

Figure 5: For a `constrained packing` task of picking-and-placing all the boxes on the rack: Task agnostic secondary placement objectives help in realizing accurate and consistent state-action sequences in **Top** Simulation and **Bottom** open loop hardware rollouts.

**Generalizing to unseen sequences.** We next highlight the generalization ability of GSC by evaluating it on more complex unseen task plans. First, we increase the maximum horizon dependency of action by changing the goal to position not only the box but also the hook in such a fashion which complies with the success of the "push" skill in the `rearrangement push` domain. The resulting

Table 2: The success rate of GSC algorithm with and without task-agnostic constraint handling is shown for `Constrained Packing` task and compared with relevant baselines.

| Methods | Constrained Packing | | |
|---|---|---|---|
| | Task 1 | Task 2 | Task 3 |
| Random CEM | 0.45 | 0.45 | 0.10 |
| DAF (Generalization) | 0.45 | 0.70 | 0.0 |
| STAP (Policy CEM) | 0.65 | 0.68 | 0.20 |
| GSC (Ours w/o constraint guidance) | 0.70 | 0.80 | 0.50 |
| GSC (Ours w/ constraint guidance) | 0.75 | 1.0 | 1.0 |

state sequence is shown in Figure 1 (bottom). Secondly, we execute our algorithm in a closed-loop fashion and use the skill success classifier to indicate sub-task completion. The sampled plan is executed and the resulting state is checked for subsequent skill feasibility. In case, the state satisfies the pre-condition for a future or previous skill, the plan is resampled from that skill to complete the task. We demonstrate this perturbation experiment on a Franka Panda arm (video attached to supplementary, hardware execution details in Appendix H).

**Importance of the forward and backward dependency.** The dependency variable $\gamma$ governs the flow of information from the initial state (forward) and the goal (backward) during the diffusion process. We provide a qualitative study of this feature in Figure 6. In the case where the forward flow information is weak ($\gamma = 0$), the model tends to hallucinate and predict states that are inconsistent with the initial state. When the backward flow is weak ($\gamma = 1$), the model becomes myopic and fails to solve the task. We empirically found that $\gamma = 0.5$ achieves a balanced performance.

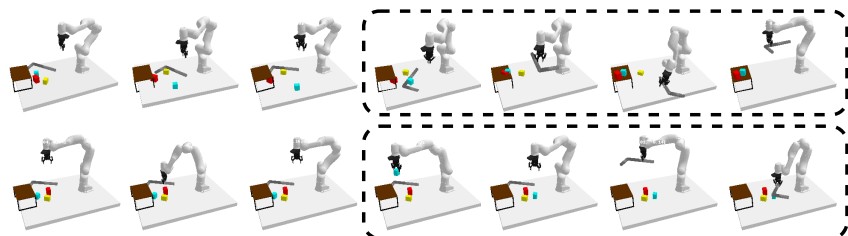

Figure 6: Example of `rearrangement push` task rollout with different dependency factor $\gamma$: **Top: Hallucination** ($\gamma = 0$) With a weak forward signal, the framework fails to realize correct position of objects at the final stage. **Bottom: Myopic** ($\gamma = 1$) A weak backward information behaves like policy shooting and fails to understand long-horizon dependency.

## 6 Limitations

The proposed framework considers planning with a given skeleton. While we do not solve the complete TAMP problem, our method is compatible with any skeleton-planning method and hence is a crucial segment of a unified framework for solving TAMP problems. Further, we only validated on a fully observable environment setup with no degree of partial observability and operates on low-dimensional state space of the system i.e. 6-DoF poses of the objects. We use a fixed set of primitive skills, and thus, the framework requires either expert data to train models or the pre-trained models to perform compositional planning. This can be extended by incorporating skill discovery frameworks.

## 7 Conclusion

We introduced GSC, a new paradigm to solve TAMP tasks with given skeletons using skill-centric diffusion models. GSC trains high-quality skill diffusion models using a transformer backbone and composes skeleton-specific distributions for unseen skeletons by chaining trained individual skill distribution. Such skeleton-specific distributions are then used to generate long-horizon paramterized skill plan sequences. The framework is scalable and flexible and shows better constraint-handling capacities, and generalizes well to new scenarios, including perturbations and replanning.

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

# A  Summary of the Algorithm

---

**Algorithm 1:** Generative Skill Chaining (GSC) Algorithm

---

1  **Hyperparameters:**
2  Number of reverse diffusion steps $T$
3  Forward-backward dependency factor $\gamma$
4  Gradient score function weight $\alpha$

5  **Inputs**:
6  Pre-defined skill library $\Pi = \{\pi_1, \pi_2, \ldots, \pi_M\}$
7  Individual skill diffusion score functions $\epsilon_\pi$
8  Task skeleton $\Phi = \{\pi_0, \pi_1, \ldots, \pi_K\}$ be a sequence of skills of length $K$
9  Initial state $\mathbf{s}^{(0)}$
10  Goal condition $g$
11  Constraints $h(\{\mathbf{s}, \mathbf{a}\})$: suppose $\{\mathbf{s}, \mathbf{a}\} = [\mathbf{s}^{(1)}, \mathbf{a}^{(2)}, \mathbf{s}^{(K)}]$ be the nodes affected by constraint

12  Initial skeleton solution $\mathbf{x}_T = [\mathbf{s}_T^{(0)}, \mathbf{a}_T^{(0)}, \mathbf{s}_T^{(1)}, \mathbf{a}_T^{(1)}, \ldots, \mathbf{s}_T^{(K)}]$ sampled from $\mathcal{N}(\mathbf{0}, \mathbf{I})$
13  Initialize $t = T$
14  **while** $t \geq 0$ **do**

15      // Score of skeleton sequence
16      $\epsilon_\Phi(\mathbf{s}_t^{(0)}, \mathbf{a}_t^{(0)}, \mathbf{s}_t^{(1)}, \mathbf{a}_t^{(1)}, \ldots, \mathbf{s}_t^{(K)}, t) = \mathbf{0}$
17      **for** $i = 1 : K$ **do**
18          // Update subvectors of $\epsilon_\Phi$
19          $\epsilon_\Phi(\mathbf{s}_t^{(i-1)}, \mathbf{a}_t^{(i-1)}, \mathbf{s}_t^{(i)}, t) = \epsilon_\Phi(\mathbf{s}_t^{(i-1)}, \mathbf{a}_t^{(i-1)}, \mathbf{s}_t^{(i)}, t) + \epsilon_{\pi_i}(\mathbf{s}_t^{(i-1)}, \mathbf{a}_t^{(i-1)}, \mathbf{s}_t^{(i)}, t)$
20          $\epsilon_\Phi(\mathbf{s}_t^{(i)}, t) = \epsilon_\Phi(\mathbf{s}_t^{(i)}, t) - \left( \gamma \epsilon_{\pi_i}(\mathbf{s}_t^{(i)}, t) + (1 - \gamma)\epsilon_{\pi_{i+1}}(\mathbf{s}_t^{(i)}, t) \right)$
21      **end**

22      // Constraint handling
23      **for** $v \in [\boldsymbol{s}^{(1)}, \boldsymbol{a}^{(2)}, \boldsymbol{s}^{(K)}]$ **do**
24          // Update subvectors of $\epsilon_\Phi$
25          $\epsilon_\Phi(\mathbf{v}_t, t) = \epsilon_\Phi(\mathbf{v}_t, t) - \alpha \nabla_{\mathbf{v}_t} \log h(\tilde{\mathbf{s}}^{(1)}, \tilde{\mathbf{a}}^{(2)}, \tilde{\mathbf{s}}^{(K)})$
26      **end**

27      // Obtain denoised samples
28      $\tilde{\mathbf{x}}_0 = \mathbf{x}_t + \sigma_t \epsilon_\Phi(\mathbf{s}_t^{(0)}, \mathbf{a}_t^{(0)}, \mathbf{s}_t^{(1)}, \mathbf{a}_t^{(1)}, \ldots, \mathbf{s}_t^{(K)}, t)$

29      // Get the updated noisy samples
30      $q_{0(t-1)}(\mathbf{x}_{t-1}|\tilde{\mathbf{x}}_0) = \mathcal{N}(\mathbf{x}_{t-1}; \tilde{\mathbf{x}}_0, \sigma_{t-1}^2 \mathbf{I})$
31      $t = t - 1$
32  **end**
33  Return $\mathbf{x}_0$

---

**Hyperparameters and Computation**  The number of reverse diffusion timesteps is an important parameters which plays a key role in deciding the time required to complete the sampling while keeping up with the quality of the generated samples. While a lower number of steps reduces the time taken for sampling, higher number of steps leads to finely denoised high-quality samples. We try with numerous values (256, 128, 64, and 50) and converge to using 128 diffusion steps for most of the tasks. The dependency factor $\gamma$ is set to be $0.5$ following the explanations described in section 4 and section 5 ( Figure 6). A value of $\gamma = 1$ makes GSC the same as a trivial policy rollout approach. Finally, in case of gradients, we finetune the weights to balance the effect of constraints in the reverse diffusion process. While it is difficult to drastically change the sampling trajectory due to the intricacies of the reverse process, we use $\alpha = 1$ for all our tasks with given planning constraints.

**Individual Diffusion Model Score Function Hyperparameters**

We follow the score-network architecture of DiT [44] and adopt to their open-source implementation: `github.com/facebookresearch/DiT`. We use the following hyperparameters for building our score-network:

Table 3: Hyperparameters for Score-Network with Transformer Backbone

| Hyper-parameter | Value |
|---|---|
| Hidden Dimension | 128 |
| Number of Blocks | 4 |
| Number of Heads | 4 |
| MLP Ratio | 4 |
| Dropout Probability | 0.1 |
| Number of Input Channels | 17 |
| Number of Output Channels | 17 |

# B    Additional Discussion

**Implementation details.** The performance of the proposed skill sequencing framework depends on the diversity of the expert dataset, the maximum horizon dependency of action in the unseen skeleton, and the quality of the trained skill diffusion models. Furthermore, only the true distributions are estimated and used to sample candidate solutions. To ensure high-success probability for all our tasks, we consider sampling multiple candidate sequence solutions (two of them are shown in Figure 7) and consider the best probable solution based on the product of individual skill success probability metric.

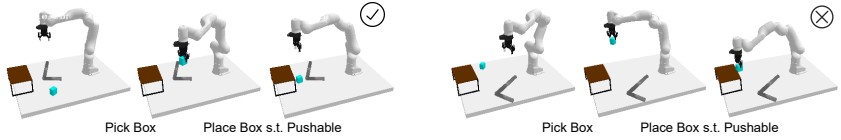

Figure 7: For a "pick-place" task of picking-and-placing the cyan box such that it can be pushed inside the rack: **Left** A correctly sampled state sequence while **Right** is incorrect. Hence, filtering candidate solutions is necessary.

**Decision diffuser approach: state diffusion model with inverse dynamics actions.** Diffusion models have been used for planning in robotics. One such framework is that of the decision diffuser [40], which samples a desired state trajectory and uses an inverse dynamics model to find the best action sequence. Our framework can achieve this by removing the action from the samples. However, this results in the distribution generated by diffusion models to be disjoint from the actions given by the inverse dynamics models. This distribution shift is sensitive to the quality of the sampled states and hence results in cascading errors. Considering a joint distribution of *state-action-state* transitions is advantageous as it is less sensitive to such state perturbations.

# C   Toy Domain: Explained

(A) Consider a toy domain with four different state (2D points) distributions and skills represented by unit vectors.

(B) We train two diffusion models:

1. One with transitions from red square to two circles. This is analogous to training a "pick hook" skill which is a pre-condition for both "pull" and "push".

2. Other with the transition from bottom circle to green square. This is analogous to training only "push" (or "pull")

Now we merge both of these transitions to sample candidate solutions for the skeleton (pick hook, push block). When sampled parallely, the first diffusion model (pick skill) will sample post-conditions from both the circles (push and pull) whereas the second diffusion model (push) will sample pre-conditions only from the bottom circle. So, eventual outcome will be from red square to bottom circle to green square (state sequence where hook is picked such that block can be pushed).

(C) We then impose conditions of "fixed initial state" and sample the chain.

(D) We then limit the goal states by imposing conditions that goal state variables must lie on the left half of the green square distribution.

(E) We then fix both initial and goal states. This samples a single solution. (as the states, 2D points, are connected via unit vectors with direction as the skill parameter)

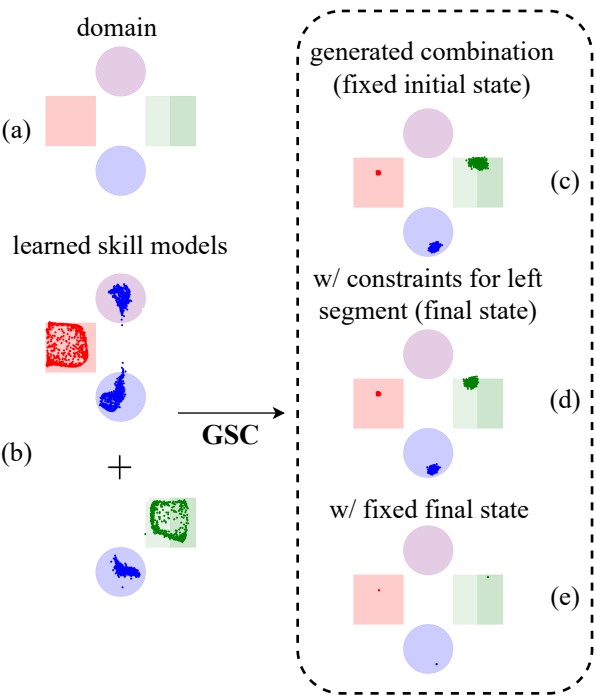

Figure 8: **Toy Domain:** We model four distributions of states and segment one of them into left and right segments. The above figure illustrates diffusion model composition using GSC with fixed start state and unit actions followed by the addition of soft and hard constraint guidance.

# D   Skill Descriptions and Parametrization

We have a fixed library of skills consisting of: `pick`, `place`, `push` and `pull`. Each of the skills is parameterized with respect to the objects of interest according to the following setup:

**pick**: Parameterized by $(x, y, z, \theta)$ as the pick location and the gripper's orientation around the z-axis. The parameters are calculated with respect to the object of interest's (to be picked) origin. For example, `pick block` is w.r.t. the origin of the block, which is the centroid. Similarly, for `pick hook`, the origin is the center of the rectangle, of which the hook is one L-segment.

**place**: Parameterized by $(x, y, z, \theta)$ as the place location and the gripper's orientation around the z-axis. The parameters are calculated with respect to the object of interest's (on which the picked object will be placed) origin.

**push**: Parameterized by $(x, y, \theta, r)$ as the location and orientation of the placement of the tool (hook) on the table ($z = 0$) and $r$ denotes the length by which the hook will be displaced away from the arm base. The parameters $x, y, \theta$ are w.r.t. the object of interest's (to be pushed) origin. The push distance $r$ is the position displacement of the tool in direction $\theta$.

**pull**: Parameterized by $(x, y, \theta, r)$ as the location and orientation of the placement of the tool (hook) on the table ($z = 0$) and $r$ denotes the length by which the hook will be displaced towards the arm base. The parameters $x, y, \theta$ are w.r.t. the object of interest's (to be pulled) origin. The pull distance $r$ is the position displacement of the tool in direction $\theta$.

An example of the pre-condition and effect of the above skills are shown in Figure 3 (left).

# E  Task Descriptions

As described in section 5, we evaluate our framework on three task domains (`hook reach`, `constrained packing`, and `rearrangement push`) with three tasks each. In addition, we validate the algorithm on a more complex skill with longer-horizon action dependency and describe it as the fourth task under the domain of `rearrangement push`. Each of the considered suites focuses on understanding long-horizon success of one particular skill. For example, `hook reach` is about the long-term effect of executing `hook`, while `constrained packing` focusses on `place` and `rearrangement push` focuses on `push`. Each task's challenge is directly proportional to the long-horizon action dependency required to complete it. For example, `pull` affects immediately if the next skill is `pick`. But `place` affects the next skill after executing one intermediate skill (like `pick`). Similarly, action dependency is after two skills for `rearrangement push` take 4. We describe all of such considered tasks below.

**Hook Reach Task 1** sub-sequence of Figure 9

- **Scene:** Box is out of workspace, Hook is inside workspace
- **Goal:** Pick the Box
- **Skeleton: Pick** Hook, **Pull** Box, **Place** Hook, **Pick** Box

**Hook Reach Task 2** easy version of Figure 9

- **Scene:** Yellow Box is out of workspace, Blue Box inside the workspace, Hook is inside workspace, Rack is inside workspace, Rack is empty
- **Goal:** Yellow Box on Rack
- **Skeleton: Pick** Hook, **Pull** Yellow Box, **Place** Hook, **Pick** Yellow Box, **Place** Yellow Box on Rack

**Hook Reach Task 3** shown in Figure 9

- **Scene:** Red Box is out of workspace, Hook is inside workspace, Rack is inside workspace, Rack already has two blocks (Yellow and Blue)
- **Goal:** Red Box on Rack (without collision)
- **Skeleton: Pick** Hook, **Pull** Red Box, **Place** Hook, **Pick** Red Box, **Place** Red Box on Rack

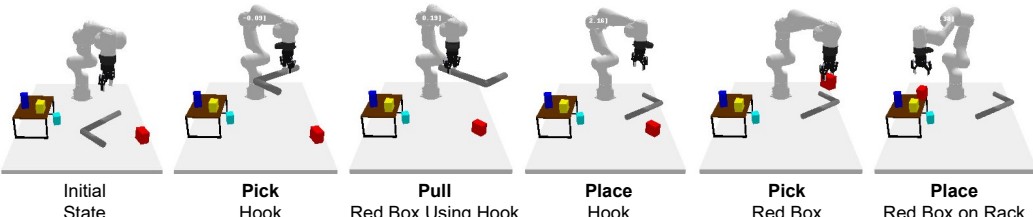

| Initial State | **Pick** Hook | **Pull** Red Box Using Hook | **Place** Hook | **Pick** Red Box | **Place** Red Box on Rack |

Figure 9: Hook Reach Task 3

**Constrained Packing Task 1** shown in Figure 10

- **Scene:** Three boxes in the workspace, Rack is in workspace, Blue block on Rack
- **Goal:** All Boxes on Rack (without collision)
- **Skeleton: Pick Box, Place** Box on Rack, **Pick** Box, **Place** Box on Rack, **Pick** Box, **Place** Box on Rack

**Constrained Packing Task 2** sub-sequence of Figure 11

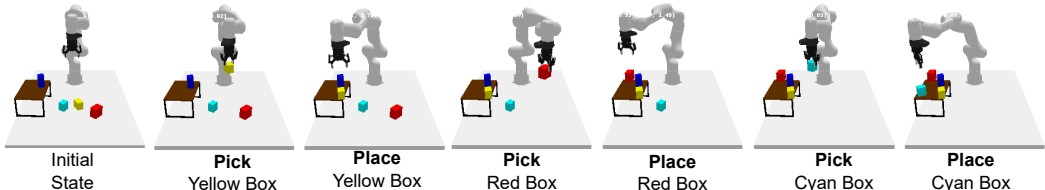

Figure 10: Constrained Packing Task 1

- **Scene:** Three boxes in the workspace, Rack is in workspace, Rack is empty
- **Goal:** Three Boxes on Rack (without collision)
- **Skeleton: Pick** Box, **Place** Box on Rack, **Pick** Box, **Place** Box on Rack, **Pick** Box, **Place** Box on Rack

**Constrained Packing Task 3** shown in Figure 11

- **Scene:** Four boxes in the workspace, Rack is in workspace, Rack is empty
- **Goal:** Four Boxes on Rack (without collision)
- **Skeleton: Pick** Box, **Place** Box on Rack, **Pick** Box, **Place** Box on Rack, **Pick** Box, **Place** Box on Rack, **Pick** Box, **Place** Box on Rack

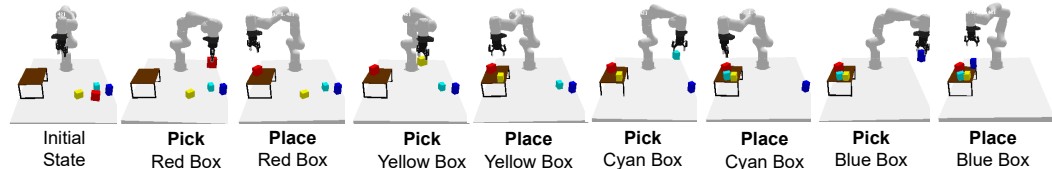

Figure 11: Constrained Packing Task 3

**Rearrangement Push Task 1** shown in Figure 12

- **Scene:** Box in workspace, Hook in workspace, Rack outside workspace
- **Goal:** Box under Rack
- **Skeleton: Pick** Box, **Place** Box, **Pick** Hook, **Push** Box using Hook

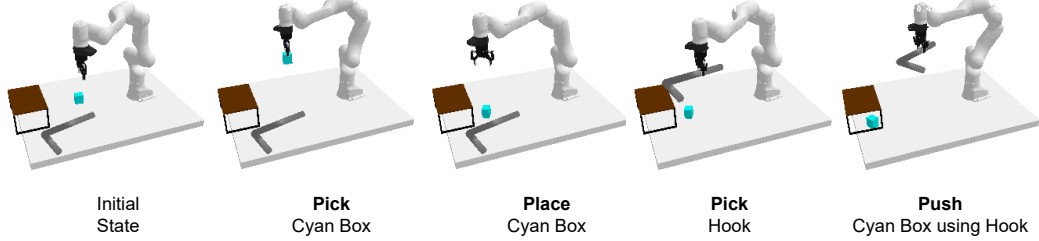

Figure 12: Rearrangement Push Task 1

**Rearrangement Push Task 2** shown in Figure 13

- **Scene:** Three Boxes in workspace, Hook in workspace, Rack outside workspace
- **Goal:** Yellow Box under Rack
- **Skeleton: Pick** Hook, **Place** Hook, **Pick** Cyan Box, **Place** Cyan Box, **Pick** Hook, **Push** Yellow Box using Hook

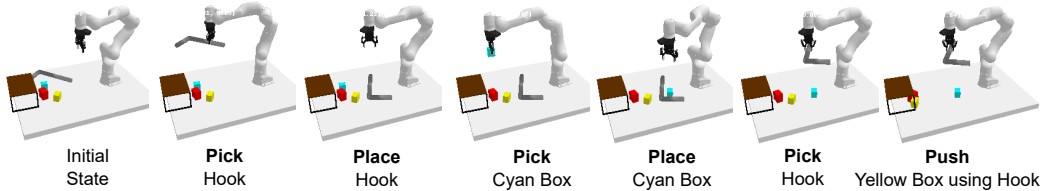

Figure 13: Rearrangement Push Task 2

**Rearrangement Push Task 3** shown in Figure 14

- **Scene:** Four Boxes in workspace, Hook in workspace, Rack outside workspace
- **Goal:** Blue Box under Rack
- **Skeleton: Pick** Red Box, **Place** Red Box, **Pick** Yellow Box, **Place** Yellow Box, **Pick** Cyan Box, **Place** Cyan Box, **Pick** Hook, **Push** Blue Box using Hook

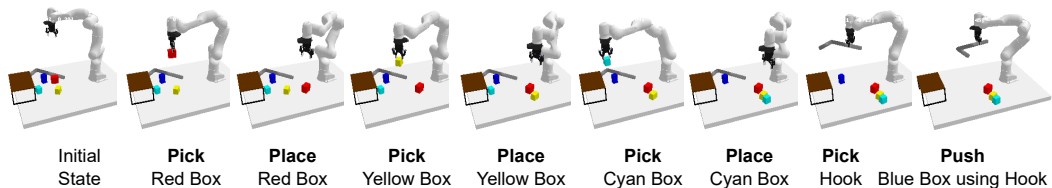

Figure 14: Rearrangement Push Task 3

**Rearrangement Push Task 4** shown in Figure 15

- **Scene:** Box outside workspace, Hook in workspace, Rack outside workspace
- **Goal:** Box under Rack
- **Skeleton: Pick** Hook, **Pull** Box using Hook, **Place** Hook, **Pick** Box, **Place** Box, **Pick** Hook, **Push** Box using Hook

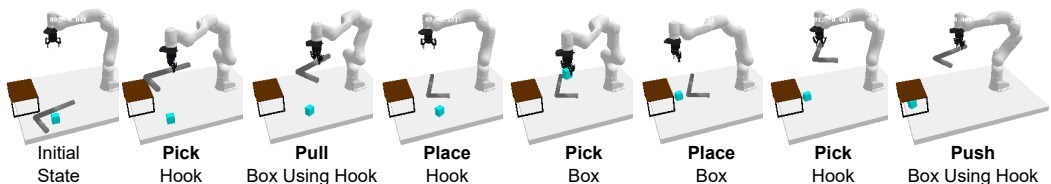

Figure 15: Rearrangement Push Task 4

## F    Additional Results

One of the attractive aspects of diffusion models is to visualize convergence to a valid solution starting from Gaussian noise. We visualize such results and show one of them below.

$t = 50$

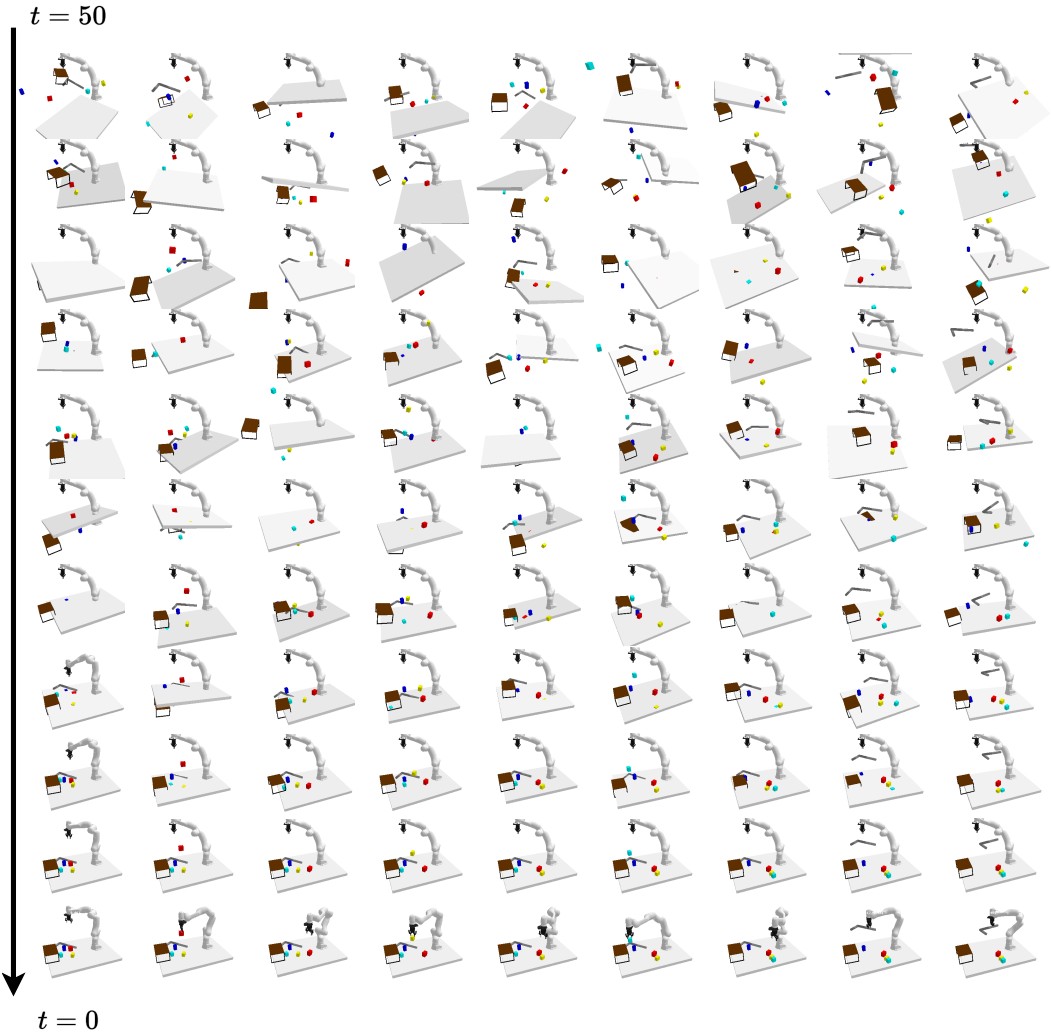

$t = 0$

Figure 16: Reverse Diffusion visualization of `Rearrangement Push Task 3` for $50$ timesteps.

## G  Hardware Experiment Setup

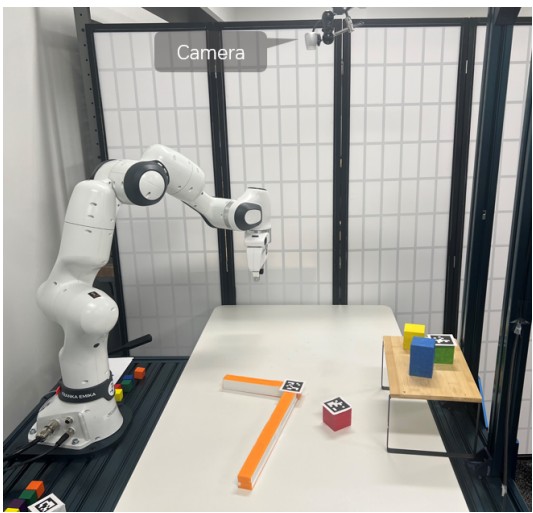

Figure 17: Hardware Experiment Setup

The experimental setup, illustrated in Fig. 17, encompasses a Franka Panda robot arm and an Intel RealSense camera, several blocks, a rack, and a hock. The camera is positioned overhead, facing downward to fully observe the poses of all the objects. AprilTag [61] is employed for SE(3) pose detection of the objects. During planning, the Frankx controller is utilized to generate smooth linear motion toward the desired gripper pose.

## H  Data Collection and Real-World Experiment Details

We collect transition data in simulation from a random agent. For every selected skill ("pick", "place", "push" and "pull"), we start with a suitable state satisfying its pre-condition and collect successful skill parameters (from random samples) and effect (resulting next state). This is done for scenes containing a varied number of objects. We collect around 5000 successful transitions from the simulator. The data is used to train the diffusion model. The success probability prediction module $Q(s', s)$ is also trained on the same data, but we add the failure transitions as well.

For real-world experiments, we use pose detection using AprilTag [61] followed by a real-to-sim scene reconstruction. All the experiments are performed with pre-trained diffusion models (trained on simulated data). The closed-loop planning in the real world is performed in the reconstructed scene in the simulator, and planned skill parameters are executed directly in the real world. The scene is updated before each replanning phase.

