# OpenReview forum: "Generative Skill Chaining: Long-Horizon Skill Planning with Diffusion Models"
_robot-learning.org/CoRL/2023/Conference — CoRL 2023 Poster_

### Official Review · Reviewer_oxXV · 2023-07-13

**Confidence:** 4
**Originality:** Very Good
**Technical Quality:** Very Good
**Clarity Of Presentation:** Good
**Impact:** 4

**Recommendation:**

Weak Accept: I recommend accepting the paper, but will not argue for my recommendation if the majority of other reviewers have a different opinion.

**Review:**

Strengths

- The range of experiments in the paper and the results are impressive. The demonstration on real hardware is particularly impressive.

- While the potential composability with diffusion models has been quite well explored in prior work, notably in [40] as cited by the authors but also in [A, B] (which should be cited), the authors have applied these techniques in a way that is particularly relevant to robotic manipulation planning and I believe is novel.

Weaknesses:
- There is a mistake in both equations under line 151, line 156 and line 157, which makes things unclear. Either the distributions on the LHS should be $p_\phi(s^{0:2}, a^{0:2}|s^0)$ or the RHS should be marginalizing out $a^0$ and $a^1$. It seems that it is probably the former, given the resulting score functions in equations (3) and (4).

- There is quite a lot of detail missing from the paper which would make reproduction difficult. For instance, while the skills are described in Appendix C, what are their parameterizations and how are they executed? How is the expert data generated for learning the diffusion models? How much data is collected for each? How is the success probability prediction module Q(s', s) trained and does it use the same data as the diffusion models? Are the real-world demonstrations performed with data collected in simulation or in the real world?

- In several places the authors claim to be solving TAMP problems. Given that the skill sequence is given as an input, I don't believe it is reasonable to claim the method solves TAMP problems. If the proposed method can be used to efficiently generate skill sequences for a given candidate skill skeleton then it would very interesting to explore it's use in a system which solves the full TAMP problem, but this is out of scope of the current paper, so it would be better to revise the claims.

Additional questions:
- How long does it take to sample from the diffusion model?
- What proportion of the generated skill sequences are feasible? How many skill sequences need to be generated to reliably ensure the sequence with the highest cumulative probability of success is feasible?

[A] Du, Y., Durkan, C., Strudel, R., Tenenbaum, J.B., Dieleman, S., Fergus, R., Sohl-Dickstein, J., Doucet, A. &amp; Grathwohl, W.S..,  Reduce, Reuse, Recycle: Compositional Generation with Energy-Based Diffusion Models and MCMC, ICML 2023

[B] Liu, N., Li, S., Du, Y., Torralba, A., Tenenbaum, J.B., Compositional Visual Generation with Composable Diffusion Models, ECCV 2022



**Quality Of The Limitations Section:**

Limitations are addressed clearly

**Questions For Rebuttal:**

- Fix mistakes in equations under line 151, line 156 and line 157
- Add missing details as outlined above

**Robotics Focus:**

Sufficient demonstration on hardware

**Summary Of Paper:**

This paper proposes a learning-based method for long horizon planning using sequences of pre-defined skills. The authors propose learning generative models of the state/skill parameter distributions under each skill, training a seperate diffusion model for each skill. The authors propose a method for composing the diffusion models at test time in order to sample feasible sequences of states & skill parameters to solve unseen tasks. The authors demonstrate their approach both in simulation and in real hardware showing improvement over baselines.

**Summary Of Recommendation:**

Overall I believe this is an interesting paper which is very relevant to the CoRL community and has impressive experimental results. However the paper has several issues that I have outlined above which if fixed, would result in a strong paper.

---

### Official Review · Reviewer_1DYZ · 2023-07-19

**Confidence:** 4
**Originality:** Very Good
**Technical Quality:** Very Good
**Clarity Of Presentation:** Excellent
**Impact:** 3

**Recommendation:**

Weak Accept: I recommend accepting the paper, but will not argue for my recommendation if the majority of other reviewers have a different opinion.

**Review:**

Strengths
- The high-level idea of chaining skills by sampling  a sequence of generative models in parallel is very natural and reasonable. By modeling the joint distribution of precondition, controls and effect, planning is framed as sampling from a generative model. The compositionality of the method is impressive, i.e., it can generalize to new plan skeletons without any fine-tuning. It's also pretty flexible, i.e., constraints can be incoporated easily through classifier-based guidance. Additionally, it's amenable to replan.


Weaknesses
- The paper assumes that the plan skeleton is given, i.e., the sequence of skills that the robot should use, which is a very strong assumption. The authors claim that the proposed method is able to generate long-horizon plan, but in reality, it's generating the low-level controls of each skill. I believe knowing which skill to use is also very important and the authors should at least mention it in the limitation section.
- The tasks shown in the paper are a bit simple. Diffusion model is known for its great capability to model complex distributions, such as images and videos. It's a waste to use it for modeling actions like pushing and pulling.
- I also have some concerns over the scalability of the method. Since the diffusion models only operate locally, the time required for the goal to be propogated to the start of the trajectory grows linear wrt the length of the skeleton. Therefore, I believe parallel sampling from multiple diffusion models is much harder than sampling from a single diffusion model, especially when the sequence is long. I'm concerned that it may require more sampling steps and even fail to obtain realistic sample.

**Quality Of The Limitations Section:**

Additional details required

**Questions For Rebuttal:**

- I'm not sure if I fully agree that the test-time optimization based methods are discriminative while GSC is generative, and that's a stark difference. On one hand, you can see optimizing the feasibility of a given plan as optimizing the likelihood of the trajectory, or generate a trajectory that has high likelihood. On the other hand, the underlying generative model of GSC, diffusion model, is based on an iterative sampling strategy that can be seen as optimization with learned gradient.

- Why do you train a unconditional diffusion model for each skill instead of training a single skill-conditioned diffusion model?

- As mentioend above, I have some concerns about sampling in parallel over a long sequence of skills. Does it guarantee to generate samples that follows all the factorized distributions? Does it require more steps than it normally is and how much does the length of the sequence affect this? It would be great if the authors can show theoretically or empirically.

- During the skill sequencing phase, is it necessary to separate the forward and backward feasibility? According to the results, it seems that treating them equally achieves the best performance, which is expected.

-  It'll be nice to show the length of plans for each tasks in table 1, so the readers can get a sense of the difficulties of the task.

**Robotics Focus:**

Sufficient demonstration on hardware

**Summary Of Paper:**

The paper introduces Generative Skill Chaining (GSC) , a skill chaining method for long-horizon manipulation task. GSC is a probabilistic framework that learns skill-conditioned diffusion models and composes their learned distributions based on a predefined skill skeleton. More specifically, the proposed method captures skill's joint distribution of precondition, control parameters, and effect using diffusion model. A complete plan is recovered by the parallel sampling of multiple skill-centric diffusion models.
GSC has the following advantages:
1. It doesn't require knowing the true system state and dynamics model of the environment.
2. It's able to generalize to unseen tasks.
3. Robustness towards perturbation by replanning.

The effectiveness of the method is demonstrated on both simulated and physical long-horizon manipulation tasks.

**Summary Of Recommendation:**

The paper presents a framework to solve skill chaining with generative modeling. Although I have several questions regarding the details of the paper, I think it contains interesting ideas that the community can benefit from. Thus, I recommend accepting the paper.

---
Although it's interesting to see skill-chaining can be attained by sampling multiple diffusion model in parallel, it seems the that method struggles when horizon is long (8 steps). After reading the response from the authors, I would like to stay with my original rating.

---

### Official Review · Reviewer_VAfs · 2023-07-20

**Confidence:** 3
**Originality:** Very Good
**Technical Quality:** Very Good
**Clarity Of Presentation:** Good
**Impact:** 4

**Recommendation:**

Weak Accept: I recommend accepting the paper, but will not argue for my recommendation if the majority of other reviewers have a different opinion.

**Review:**

- The idea of combining skill chaining with diffusion models is novel and makes sense. The use of diffusion models brings better generalization of skill chaining and more flexibility on additional constraints.

- The paper is generally well written but some additional details in the experiments would be appreciated.

- The experiments look solid at a high level but the details are not very clear to me. It would be helpful if the paper describes what each task suite tries to show, why it is challenging, and why things succeed or fail.

**Quality Of The Limitations Section:**

Limitations are addressed clearly

**Questions For Rebuttal:**

- It is not easy to understand Figure 4. A bit more explanation in the caption (e.g. what do rectangles and circles mean? what does color represent? what is skill and what is state?) or labels in the figure would be helpful to understand it.

- In experiments, it is not clear how to collect transition data. Is it granted? or collected from an expert? or with pre-defined skills? Then, what are the skills? How are they defined? It might be trivial from the prior work but for those who are not knowledgeable in the prior work, it would be great if the paper explains it even at a high level.

- In the constrained packing results in Figure 5 and Table 2, it is not clear to me what are the additional constraints imposed on the task.

- Similar to the generalization experiments, it would be great if more detailed information is provided about what task plans are used for training and what are used for testing, and what types of generalization are expected from these experiments. Also, some quantitative results for generalization would be helpful.

**Robotics Focus:**

Sufficient demonstration on hardware

**Summary Of Paper:**

This paper proposes to find skill parameters for skill chaining using a diffusion model given a skill sequence (skeleton) of a target task and skill policies. Thanks to diffusion models, the proposed approach is flexible to additional constraints in any intermediate states or skill parameters, so that a combination of an initial state, intermediate state, final state, and skill behavior can be easily specified for planning. The experiments show that the proposed method works well under various planning scenarios and shows improved success rates.

**Summary Of Recommendation:**

The proposed approach of combining diffusion models and skill chaining is interesting and novel. The proposed method makes planning with new constraints easy, which makes skill chaining much more flexible for various scenarios. I would recommend weak acceptance for this paper.

----

The analysis and presentation of the paper could be improved to support the claim of this paper. I would stay with weak accept.

---

### Decision · Program_Chairs · 2023-08-30

**Decision:**

Accept (Poster)

**Comment:**

This paper proposes a method for chaining skills based on diffusion. The method takes in a sequence of skills for a target task and skill policies. A plan is formed via sampling from composed diffusion models. The approach is flexible in that it can incorporate constraints on intermediate states or skill parameters. The results demonstrate that the method performs well in several scenarios.

Strengths:
- The use of diffusion models allows generalization of skill chaining to additional constraints
- Not requiring foreknowledge of the environment dynamics is an advantage
- The method can respond to perturbations via replanning

Weaknesses:
- The details of the experiments are not very clear (see review comments for specific questions)
- The paper assumes that the plan skeleton is given, but this is not often the case for real-world tasks
- The tasks shown in the paper are not very complex, more challenging tasks should be considered
- Any claims related to solving TAMP problems should be removed, as there is no task-level planning (the plan skeleton is given)

Overall, the rebuttal did not significantly change the reviewers' opinions.